# Knowing When to Silence: Roles of Polycomb-Group Proteins in SAM Maintenance, Root Development, and Developmental Phase Transition

**DOI:** 10.3390/ijms21165871

**Published:** 2020-08-15

**Authors:** Bowen Yan, Yanpeng Lv, Chunyu Zhao, Xiaoxue Wang

**Affiliations:** Rice Research Institute, Shenyang Agricultural University, Shenyang 110866, China; 2018200065@stu.syau.edu.cn (B.Y.); 2019200060@stu.syau.edu.cn (Y.L.); 2019220264@stu.syau.edu.cn (C.Z.)

**Keywords:** PRC1, PRC2, PcG proteins, plant development, Arabidopsis, rice

## Abstract

Polycomb repressive complex 1 (PRC1) and PRC2 are the major complexes composed of polycomb-group (PcG) proteins in plants. PRC2 catalyzes trimethylation of lysine 27 on histone 3 to silence target genes. Like Heterochromatin Protein 1/Terminal Flower 2 (LHP1/TFL2) recognizes and binds to H3K27me3 generated by PRC2 activities and enrolls PRC1 complex to further silence the chromatin through depositing monoubiquitylation of lysine 119 on H2A. Mutations in PcG genes display diverse developmental defects during shoot apical meristem (SAM) maintenance and differentiation, seed development and germination, floral transition, and so on so forth. PcG proteins play essential roles in regulating plant development through repressing gene expression. In this review, we are focusing on recent discovery about the regulatory roles of PcG proteins in SAM maintenance, root development, embryo development to seedling phase transition, and vegetative to reproductive phase transition.

## 1. Introduction

Higher eukaryotes have evolved mechanisms to temporally and specially control gene expression. Polycomb-group (PcG) proteins have been initially identified as transcriptional repressors of *Homeobox* (*Hox*) genes in *Drosophila* (*Drosophila melanogaster*) to control embryonic development, stem cell differentiation, and tissue homeostasis development [1,2]. PcG proteins usually form multiple complexes, such as polycomb repressive complex 1 (PRC1) and PRC2, to inactivate gene expression and maintain the silencing state of the target chromatin through covalent histone modifications [3]. PRC1 complexes are E3 ubiquitin ligases that modify the chromatin via depositing monoubiquitylation of lysine 119 mark on histone 2A (H2AK119ub). PRC2s are methyltransferases that target lysine 27 on histone 3 for trimethylation (H3K27me3) and generates H3K27me3 epigenetic mark on the target loci [4,5]. Most of the components in PRC1 and PRC2 complexes are conserved from *Drosophila* to mammals and plants, homologs of which have been characterized recently [6,7].

In *Drosophila*, PRC2 is composed of four core components, including the SET domain-containing histone methyltransferase enhancer of zeste [E(z)], Suppressor of Zeste 12 [Su(z)12], Extra sex combs (Esc), and Nucleosome remodeling factor 55 kDa subunit (Nurf55) or p55 [8] (Table 1).

In Arabidopsis (*Arabidopsis thaliana*), members of these core subunits are present in multigene families. CURLY LEAF (CLF), MEDEA (MEA) (also termed Fertilization Independent Seed 1, FIS1), and SWINGER (SWN) are three homologs of E(z) which have methyltransferase activity to trimethylate H3K27 of the target chromatin [9,10,11]. Embryonic Flower 2 (EMF2), VERNALIZATION 2 (VRN2), and Fertilization Independent Seed (FIS2) are the three homologs of Su(z)12, which promote the activity of H3K27 methyltransferase and confer target specificity to the corresponding PRC2s [11,12,13]. Multiple Suppressor of IRA 1 to 5 (MSI1 to MSI5) are the five homologs of Nurf55/p55, but only the functions of MSI1 and MSI4 have been characterized [14,15,16,17]. Fertilization Independent Endosperm (FIE) is the homolog of Esc in Arabidopsis [18,19,20,21]. Three different PRC2s, including EMF-PRC2, VRN-PRC2, and FIS-PRC2 complexes, have been characterized in Arabidopsis based on the homologs of Su(z)12 in the complexes [22]. Two components, FIE and MSI1, are present in all three PRC2 complexes in Arabidopsis [17,23] (Table 1).

In the rice (*Oryza sativa*) genome, two E(z)-like proteins in the rice genome have been identified termed OsCLF and OsSET1/OsiEZ1, respectively, because of their high similarity to Arabidopsis CLF and SET1 [24,25,26]. There are only two Su(z)12-like proteins, designated as OsEMF2a and OsEMF2b, which are the Arabidopsis homologs of EMF2. In contrast to the single copy of an Esc homolog, FIE in Arabidopsis, there are two Esc homologs in rice, OsFIE1 and OsFIE2. The two genes are located close to each other on chromosome 8 separated by one gene. The amino acid sequences of OsFIE1 and OsFIE2 are 72% identical [24]. One homolog of MSI1 in rice termed OsMSI1 has also been reported [27] (Table 1).

The PRC1 complex in *Drosophila* is composed of dRing/Sex comb extra (Sce), Posterior sex combs (Psc), and Polyhomeotic (Ph) sununits [7,28]. In Arabidopsis, AtRING1A/B and AtRING1B are the homologs of dRing/Sce [29]. AtBMI1A, AtBMI1B, and AtBMI1C are the homologs of Psc [29,30,31]. AtRING1A, AtRING1B, AtBMI1A, and AtBMI1B possess the E3 ligase activities to monoubiquitylate H2AK119. Like Heterochromatin Protein 1/Terminal Flower 2 (LHP1/TFL2), performing as the reader of histone modification, recognizes and binds to H3K27me3 generated by PRC2 activities [32,33]. In addition, plant-specific proteins, including Embryonic Flower 1 (EMF1), VERNALIZATION 1 (VRN1), and VP1/ABI3-Like 1/2/3 (VAL1/2/3), are involved in PRC1 function [34,35,36]. However, no functional homolog of Ph has been identified in Arabidopsis genome. LHP1 integrates into PRC1 complex via interacting with both AtBMI1 and AtRING1 [29,37,38]. In addition to the involvement in PRC1 complex, LHP1 is also associated with PRC2 through interacting with MSI1 and EMF2 to deposit H3K27me3 mark on the chromatin [17,39]. In rice, a homolog of EMF1, AtRING1A/1B, and LHP1 in rice designated as Curved Chimeric Palea 1 (CCP1) or OsEMF1, OsRING1, and OsLHP1, respectively, have been reported [27,40].

During the life cycle, plants undergo several important phase transitions, from gametophytic to sporophytic stage; from embryogenesis to germination, from juvenile to adult vegetative phase, and from vegetative to reproductive phase [41]. PcG proteins play pivotal roles in controlling these processes. Whole genome data have shown that about 20% of Arabidopsis genes are marked by H3K27me3 [32,42,43]. In this review, we are focusing on recent findings on roles of PcG proteins in SAM maintenance, root development, seed to germination phase transition, and floral transition.

## 2. Roles of PcG Proteins in SAM Maintenance and Differentiation

Generation of new organs during life cycle of plants is dependent on the maintenance and differentiation of stem cells [42,44,45]. Root apical meristem (RAM) is involved in generation of underground architecture. Shoot apical meristem (SAM), axillary meristems (AMs), inflorescence meristem (IM), and floral meristem (FM) generate aboveground organs [46].

In Arabidopsis, SAMs are regulated by a negative feedback loop comprising the stem cell-promoting transcription factor WUSCHEL (WUS) and differentiation-promoting peptide CLAVATA 3 (CLV3) [47,48,49]. WUS, a homeodomain transcription factor, is required for maintenance of stem cell identity [50,51,52,53]. WUS activates CLV3 expression [50,54,55,56]. CLV1 (a receptor kinase), CLV2 (receptor like protein), and CLV3 (a ligand of CLV1) form a receptor kinase-signaling cascade to repress *WUS* expression, maintain appropriate meristem size, and promote meristem cells toward organ initiation [54,57,58,59,60].

These regulatory mechanisms are conserved in rice [61]. The WUS ortholog in rice termed *OsWUS* is expressed mainly in leaf margins but could not be always detected in SAM [62,63]. The functions of *Oryza sativa* WUSCHEL-Related Homeobox 4 (OsWOX4) is similar to WUS in Arabidopsis. RNAi transgenic lines of *OsWOX4* reduce the size of SAM and flatten SAM [64]. Rice Floral Organ Number 1 (FON1) and FON4 (also termed FON2) are homologs of CLV1 and CLV3, which play important roles in maintaining activities of the stem cells within the FM [65,66,67,68].

### 2.1. PRC2 Represses the Expression of WUS Directly or Indirectly

The expression of *WUS* is transcriptionally regulated by PRC2 complex [69]. The chromatin of *WUS* is marked by H3K27me3 [70]. The Polycomb Response Elements (PREs) from *WUS* locus, termed CArG boxes, have been characterized and are shown to be essential for establishing and spreading of H3K27me3 through PRC2 activity [70,71,72]. AGAMOUS (AG), a MADS domain-containing protein, binds directly to the CArG boxes located in the *WUS* coding region, which in turn recruits PRC2 complex to catalyze trimethylation of H3K27 and repress *WUS* expression [70]. Mutations in the CArG boxes of *WUS* disrupt the binding of AG to *WUS* and activate the expression of *WUS* (Figure 1a). Mutation in *AG* results in the decreased H3K27me3 level at the *WUS* locus [70].

In addition to directly repressing *WUS*, AG inactivates *WUS* to regulate stem cell activity through activating *KNUCKLES* (*KNU*) [72,73,74]. KNU, a C2H2-type zinc finger protein, represses the expression of *WUS* [72,73]. The expression of *KNU* is repressed by PRC2 [75,76]. There are three AG binding sites, termed *KNU*-PREs, in the promoter region of *KNU* (Figure 1b). *KNU* activation by AG is associated with the loss of H3K27me3 [71,72]. AG binds to the *KNU*-PREs by competing with PRC2, leading to the loss of H3K27me3 at the coding region and activation of *KNU*, further suppressing the expression of *WUS* (Figure 1c) [72,74,77,78,79].

### 2.2. PRC2 Inactivates KNOX Genes

The *Knotted 1*-*like homeobox* (*KNOX*) genes, including *Shoot Meristemless* (*STM*), *BREVIPEDICELLUS* (*BP*), *Knotted 1-Like from Arabidopsis thaliana 1* (*KNAT1*), *KNAT2*, and *KNAT6* are important to maintain the meristem activity of SAM [80,81,82]. They are localized at the shoot apex and function redundantly with STM [83]. The *KNOX* genes are expressed in the meristems. The *Asymmetric leaves 1* (*AS1*) and *AS2* genes, encoding MYB-domain and LOB-domain containing transcription factors, respectively, are expressed in the organ primordial [83]. The *KNOX* genes restrict the expression of *AS1* and *AS2* genes to organ primordia, thus preventing ectopic organ initiation. AS1 and AS2 can, in turn, repress *KNOX* gene expression [84,85]. These genes are crucial for distinguishing stem cells and organ founder cells.

It has been recently reported that H3K27 trimethylation of these cell fate-determination genes depends on FIE [86]. The expression of the genes, including *STM*, *WUS*, *KNATs*, and *WOX1*, are upregulated by *fie* mutation, along with the loss of H3K27me3 [86,87]. Mechanisms on PRC2-mediated H3K27me3 modification of *KNOX* genes have been recently reported [88,89]. AS1 physically associates with AS2, forming AS1-AS2 complex [85,90,91]. AS1-AS2 complex has DNA-binding activity and directly binds to the CWGTTD and KMKTTGAHW PREs localized in the promoters of *BP* and *KNAT2* to recruit PRC2, thus triggering H3K27me3 modification to silence their expression [88,89]. AS1 integrates into PRC2 through physically interacting with CLF and FIE, whereas AS2 is directly associated with EMF2 and recruit PRC2 complex to *BP* and *KNAT2* regions to modify the chromatin with H3K27me3 mark and silence SAM-specific homeobox domain genes in leaf tissue (Figure 1d) [88].

In addition, AS1-AS2 complex also interacts with LHP1 and recruits PRC2 to *cis*-acting element in the promoter regions of *BP* and *KNAT2* [92]. Mutations in *AS1* or *AS2* lead to reduction in H3K27me3 levels at *BP* and *KNAT2* loci. Furthermore, mutations in AS1-AS2 binding sites also result in decrease of H3K27me3 accumulation and the derepression of *BP* and *KNAT2* genes [88,92].

### 2.3. PRC1 Regulates Stem Cell Fates

PRC1 complex also plays roles in maintenance of SAM. Mutations of *AtRING1A* and *AtRING1B* lead to ectopic-meristem formation in cotyledons and leaves. The *atring1a/atring1b* double mutant plants display an enlarged SAM [29]. The class I *KNOX* genes, including *STM*, *KNAT1*, *KNAT2*, and *KNAT6*, are direct targets of AtRING1s [29,38]. H2AK119ub modifications in *KNOX* genes are significantly reduced in *atring1a/atring1b* double mutants, suggesting that AtRING1A and AtRING1B control stem cell fate by generating H2AK119ub and repressing the expression of target genes (Figure 1e). However, H3K27me3 deposition on the chromatin of *STM*, *KNAT2*, and *KNAT6* is undisturbed in *atring1a/atring1b* mutant plants [29,37]. Similarly, disruption of *AtBMI1A* and *AtBMI1B* activity decreases the level of H2AK119ub in vivo, but did not alter H3K27me3 deposition at target genes (Figure 1e) [29,31,36,37]. These results indicate that AtRING1s and AtBMI1s are required for H2AK119ub, but that they are not required for the trimethylation of H3K27 by PRC2 [29,37].

## 3. Roles of PcG Proteins in Root Development

### 3.1. PRC2 Represses ABI4 to Control Root Development

Involvement of PRC2 in root development has been characterized recently. Abscisic acid Insensitive 4 (ABI4), an AP2/ERF domain-containing transcription factor, regulates root development [93]. Disruption of *ABI4* gene increases the number of lateral roots, whereas overexpression of *ABI4* decreases lateral root density. The effect of ABI4 on lateral root development is mediated through repression of *PIN-FORMED 1* (*PIN1*), which encodes an auxin-efflux carrier and regulates auxin distribution [93]. Basic Pentacysteine (BPC) proteins have been identified as plant-specific transcription factors [94,95]. BPCs proteins contain a conserved DNA-binding domain at the C-terminus and can bind to the GAGA-*cis* elements of their target genes [96,97]. BPCs are able to recruit PRC2 through interacting with SWN to the promoter of *ABI4* to repress the expression of *ABI4* by catalyzing H3K27me3 and regulate root development [98].

### 3.2. PcG Proteins Control Root Development through NRT2.1 under Lower ${NO}_{3}^{-}$ Conditions

Nitrogen (N) is a limiting factor for plant development. Low levels of nitrate stimulate lateral root formation, whereas high levels suppress lateral root formation. In addition, different N forms such as nitrate (${\mathrm{NO}}_{3}^{-}$), ammonium (${\mathrm{NH}}_{4}^{+}$), or some amino acids are signaling molecules modulating plant development [99,100]. In Arabidopsis, *Nitrate Transporter 2.1 (NRT2.1)* encodes a key high-affinity root nitrate (${\mathrm{NO}}_{3}^{-}$) transporter, crucial for ${\mathrm{NO}}_{3}^{-}$ uptake and thus for N nutrition during lateral root development and plant growth [101]. The *nrt2.1* mutants show a dramatic reduction of growth under low and limiting ${\mathrm{NO}}_{3}^{-}$ availability. It has recently been observed that *NRT2.1* is modified by H3K27me3 [33,102]. PRC2 directly targets and downregulates *NRT2.1* in the nutritional conditions where this gene is one of the most highly expressed genes [103]. The mutation of *CLF* leads to the loss of H3K27me3 at *NRT2.1* and results in the activation of *NRT2.1* expression.

### 3.3. APOLO lncRNA Recruits PRC1 to Control Root Development

APOLO (AUXIN-REGULATED PROMOTER LOOP), a long noncoding RNAs (lncRNAs), is involved in lateral root development in Arabidopsis [104]. APOLO recognizes its targets by sequence complementarity to form DNA-RNA duplexes (R-loops). The recognition of APOLO to the target DNA recruits PRC1 through LHP1, which regulates the local chromatin three-dimensional conformation changes. APOLO lncRNA modulates the expression of distal unrelated auxin-responsive genes during lateral root development [104]. The findings are consistent with the report in *Drosophila*. R-loops are able to be formed at many PREs in *Drosophila* embryos and are correlated with repressive states of the target genes. PRC1 and PRC2 can recognize R-loops and open DNA bubbles. In addition, PRC2 activity drives formation of RNA-DNA hybrids, the key component of R-loops, from RNA and dsDNA [105].

## 4. Roles of PcG Proteins in Embryo to Seedling Phase Transition

The repressive activities of PcG proteins are required for the embryo to seedling phase transition [86,87,106]. Plants produce and store food in the seed, especially in endosperm to support germination and seedling growth. During germination, genes specifying embryogenesis, seed dormancy, and accumulation of seed storage compounds must be repressed to allow embryo to grow and differentiate into plants [107].

### 4.1. PRC2 Is Required for Transition from Embryogenesis to Seed Maturation

In Arabidopsis, as in most of the Angiosperms species, the female gametophyte (embryo sac) contains a haploid egg cell and a diploid central cell. After fertilization, the central cell forms the triploid endosperm and the egg cell generates the diploid zygote [108]. In the absence of fertilization, the autonomous endosperm development is repressed by FIS-PRC2, while seed coat development is suppressed by EMF-PRC2 and VRN-PRC2 [109,110,111]. Mutations in FIS-PRC2 or impairment of auxin synthesis and signaling affect the development of embryo, endosperm, and seed coat causing seed abortion [111,112,113].

*SEEDSTICK* (*STK*), a MADS-box gene, is specifically expressed during ovule and seed development and functions in these tissues [114,115]. During carpel development, the expression of *STK* is confined to placental tissues and ovule primordial. In mature ovules, *STK* is expressed strongly in the funiculus and integuments, which form the seed coat later [115]. Class I BPCs and the MADS-domain factors, including Short Vegetative Phase (SVP), APELATA 1 (AP1), and AGAMOUS-Like 24 (AGL24), are the key regulators of *STK*. Class I BPCs, including BPC1, BPC2, and BPC3, form homodimers and heterodimers and bind to the C-boxes in the promoter of *STK* [95,116]. SVP acts redundantly as AP1 and AGL24 to repress *STK* expression during early stages of flower development, through binding to its promoter [117,118]. Class I BPCs physically interact with SVP to repress *STK* expression in the floral meristem [117]. Class I and class II BPCs control the expression of *STK*, by depositing and/or maintaining H3K27me3 marks through interacting with LHP1 [119,120].

It has been shown that a protein homologous to HEME Activator Protein 3 (HAP3) subunits of the CAAT Box binding Factors, termed Leafy Cotyledon 1 (LEC1), and three B3-domain transcription factors, including ABA Insensitive 3 (ABI3), FUSCA 3 (FUS3), and LEC2, termed LAFL proteins, are the major factors activating the seed maturation program in a complex network [121,122,123]. The *fus3-3* mutant exhibits ovule and seed abortion [124]. The repression of *LAFL* genes during early embryogenesis has been reported [125]. *LAFL* expression is tightly controlled in specific tissues of seeds. *FUS3* is expressed during ovule development. Before fertilization, the expression of *FUS3* is restricted to the chalaza and funiculus of mature ovules. After fertilization, *FUS3* is confined in the funiculus, seed coat, and chalaza [126]. BPC1 and 2 promote the transition from ovule to seed development by repressing *FUS3* in ovule integuments. Class I BPCs interact with FIS2 and MEA in FIS-PRC2 and bind to the GA/CT *cis*-element located in *FUS3* promoter to represses the expression of *FUS3* in the integuments of mature ovules and the endosperm of developing seeds. These results suggest that BPCs-mediated repression of *FUS3* in the endosperm is necessary to coordinate endosperm and embryo growth.

### 4.2. PRC2 Regulates Seed Maturation to Germination Transition

The *LAFL* genes are sequentially expressed during seed development, but silenced during and after seed germination [127,128,129]. PRC2 is involved in the repression of these genes in vegetative tissue [106,125]. In *fie* mutant, genes functioning in late seed development, including *AGAMOUS-Like 15* (*AGL15*), *ABI3*, *FUS3*, and *LEC2*, are upregulated. The expression of genes encoding storage compounds in seeds, including *CRUCIFERIN 3* (*CRU3*), *CRUCIFERINA 1* (*CRA1*), *Seed Storage Albumin 1* (*SESA1*), *Seed Storage Albumin 2*, (*SESA2*), *Late Embryogenesis Abundant* proteins (*LEAs*), and *OLEOSINs* (*OLEOs*), are also upregulated by *fie* mutation [87,106]. It has also been shown that BPC1 interacts with EMF2 and recruit EMF-PRC2 to the promoter of *FUS3* to repress its expression during vegetative development [126].

In plants, hormones are required for regulating seed development and germination. Abscisic acid (ABA) promotes seed development, maturation, and dormancy, whereas gibberellic acid (GA) facilitates germination and seedling growth [107]. PcG proteins regulate seedling development through multiple hormone signaling pathways [86,106]. During germination, FIE and EMF1 bind to genes encoding positive regulators in ABA signaling pathway, such as *ABI3* and *ABI4*, to genes encoding negative regulators of GA signaling, e.g., *RGA-Like 3* (*RGL3*) and to genes encoding major regulatory factors that promote embryo development and maturation, such as *LEC2*, to repress their expression [86,87,106]. In *fie* mutant, positive regulator genes of ABA signaling and negative regulator genes of GA signaling, such as *ABI4*, *Delay of Germination 1* (*DOG1*), *CHOTTO 1/AINTEGUMENTA-LIKE* 5 (*CHO1/AIL5*), and *SOMNUS* (*SOM*), and *SQUAMOSA Promoter Binding Protein-Like 8* (*SPL8*), are activated because of the loss of H3K27me3 modification [86,106].

### 4.3. PRC1 Represses Seed Development Genes to Allow Germination

PRC1 is involved in seed development and seedling growth. Genes required for seed development are activated by *emf1* mutation. EMF1 binds to major seed regulated genes, such as *FUS3*, *ABI3*, and the downstream seed maturation genes, including *LEC2*, *LEA*, *OLEO2*, *Lipid Transfer Protein 3* (*LTP3*), and seed storage protein genes, which are modified by H3K27me3 [130,131]. The VAL (VP1/ABI3-LIKE) 1/2/3 proteins suppress the expression of *ABI3*, *FUS3*, and *LEC2* to initiate germination and vegetative development [121]. VAL proteins interact with AtBMI1 proteins to initiate repression of seed maturation genes. The H2AK119ub levels of the seed development genes are lower in *atbmi1a/b* than in WT seedlings. In *val1/2* mutants, the levels of H2AK119ub are also reduced at seed maturation genes, but are not affected at *WUS*, indicating that VAL proteins are specifically required for generating H2AK119ub mark at seed maturation genes [36]. The regulation of seed maturation genes by AtBMI1s does not follow the classic hierarchical model proposed for animal PcG-mediated repression [132]. The PRC1 activity is required for the H3K27me3 modification of these genes to maintain the suppression status. Whether VALs is responsible for recruiting PRC1 to the targets remain unclear.

## 5. Roles of PcG Proteins in Floral Transition

The timing of floral transition is essential for the life cycle of plants. In Arabidopsis, a facultative long-day plant, the autonomous, vernalization, photoperiod, PAF1 complexes, FRIGIDA (FRI)/FRIGIDA-Like (FRL), GA, and aging pathways form a complicated network to regulate floral transition [133,134]. Till date, key flowering time-related genes in the network have been characterized and their regulatory mechanisms have been intensively investigated. Flowering Locus C (FLC), a MADS box protein, is a central suppressor of flowering through repressing the expression of two flowering time integrators, *Flowering locus T* (*FT*) and *Suppressor of COSTANS 1* (*SOC1*) [135,136,137]. The expression of *FLC* is activated by FRI/FRL and PAF1 complexes, but repressed by vernalization and the autonomous-pathway proteins [138]. Two PRC2 complexes, VRN-PRC2 and EMF-PRC2, are involved in the inactivation of *FLC* through generating H3K27me3 modification on its chromatin [6,139,140].

### 5.1. VRN-PRC2 in Vernalization Pathway

Vernalization is the process in which flowering is accelerated by prolonged cold [141,142]. It involves PcG protein-mediated gene silencing through marking H3K27me3 modification at *FLC* chromatin [143,144,145]. The VRN2-PRC2 complex is required for vernalization-induced flowering through repressing *FLC* [13]. Vernalization Insensitive 3 (VIN3) and VRN5/Vernalization Insensitive 3-Like 1 (VIL1), two plant homeodomain (PHD) proteins, associated with PRC2 suppress the expression of *FLC* [13,140,146,147]. During vernalization, VIN3 is induced and physically interacts with VRN5/VIL1. This two PHD protein-formed heterodimer is associated with VRN-PRC2 at promoter region of *FLC* to form PHD–PRC2 complex. The PHD–PRC2 complexes deposit H3K27me3 mark at *FLC* locus during cold exposure [13,23,140,144,145,146,147,148]. After transferring to the warm condition, the PHD–PRC2 complex without VIN3 spreads across the gene resulting in high H3K27me3 across *FLC* locus, necessary for epigenetic stability through the rest of development [23,145,148]. Maintenance of this silenced state requires LHP1, suggesting the cross-talk between PRC2 and PRC1 in flowering time control [144,149].

*COLDAIR*, a sense noncoding RNA (ncRNA), is also induced after cold exposure. *FLC* is downregulated concomitantly with increased *COLDAIR* transcripts [150]. The cold-induced *COLDAIR* is involved in PRC2 recruitment [151].

### 5.2. EMF-PRC2 in Floral Transition

The EMF-PRC2 plays important roles in repressing reproductive development during vegetative growth [9,12,18,152]. Mutants of *EMF2* gene skip vegetative growth and flower upon germination [153]. EMF-PRC2 complex is attributed to the ectopic expression of flower organ identity or flower MADS box genes, such as *AG*, *APETALA1* (*AP1*), *AP3,* and *PISTILATA* (*PI*), to delay flowering [130,131,154]. In addition, EMF-PRC2 inhibits the expression of *FLC* in a vernalization independent manner, which may function in concert with Flowering Locus D (FLD) [155]. *FLD* encodes a plant ortholog of the human protein Lys-Specific Demethylase 1 (LSD1) that is involved in H3K4 demethylation [155,156]. Loss of *FLD* function causes a reduction in H3K27me3, and, conversely, loss of EMF-PRC2 activity leads to an increase in H3K4me3 [157,158]. This suggests that EMF-PRC2 and the FLD complex function coordinately to demethylate H3K4, deacetylate histones, and deposit H3K27me3 on *FLC* chromatin, leading to the establishment of a repressive chromatin environment to repress *FLC* expression. EMF-PRC2-mediated repression is an endogenous mechanism that is independent of environmental signals [159].

### 5.3. PRC1 in Floral Transition

AtBMI1C, a component of PRC1 in Arabidopsis, has been identified as flowering time accelerator. Overexpression of *AtBMI1C* promotes flowering in Arabidopsis [31]. LHP1 is a component of PRC1 through interacting with AtRING1a and acts as a reader component of PRC1, similar to *Drosophila* Pc. LHP1 displays the binding specificity to H3K27me3 through its chromodomain [29,160]. A genome-wide analysis of LHP1 showed that LHP1 is highly enriched with H3K27me3-modified loci [32,33]. The *lhp1* mutant exhibits early flowering because of the activation of the expression of flowering time-related genes *FT* and floral organ identity genes, such as *AG*, *AP3*, *PI*, and *SEPALLATA 3* (*SEP3*) [32,149,161]. In addition, LHP1 is required to maintain the epigenetically repressed state of *FLC* upon return to warm conditions [162]. To maintain *SEP3* chromatin in a silenced state, Short Vegetative Phase (SVP) interacts with LHP1/TFL2 to modulate H3K27me3 [163].

*EMF1*, another Arabidopsis gene required for vegetative development, encodes a plant-specific protein containing sequence motifs found in transcriptional regulators [164]. The *emf1* mutants have similar phenotypes to *emf2* mutants, flowering early upon germination [18,106,152,153]. Tissue-specific removal of EMF1 activity from leaf primordia allows vegetative growth, but leads to early flowering plants with curly leaves similar to *clf* mutants [165]. The expression of floral organ identity such as *AG*, *AP1*, *AP3*, and *PI* are upregulated by *emf1*, which confers its early flowering phenotype [130,131,154]. EMF1 binds to the chromatin regions of the floral organ identity genes such as *AG*, *AP1*, *AP3*, and *PI* and displays characteristics similar to PRC1 component, Psc, in *Drosophila* [35]. EMF1 is also required for Arabidopsis RING-finger protein-mediated H2AK119 monoubiquitination [37].

It has also been revealed that EMF1, LHP1, and a trimethyl histone H3 lysine-4 (H3K4) demethylase form a PcG complex, termed EMF1 complex (EMF1c) to mediate *FT* silencing in Arabidopsis. In leaf veins, EMF1c directly represses *FT* expression before the end of long days (LDs) and at night, and that in response to inductive LDs, CONSTANS (CO) accumulation at dusk antagonizes EMF1c binding to *FT* chromatin to promote FT transcriptional activation and thus the onset of flowering [159].

### 5.4. PcG Proteins in Flowering Time Control of Rice

Photoperiodic regulation is a major mechanism on flowering time control of rice. Short day (SD) promotes rice flowering. The genes involved in the photoperiod pathway are conserved between Arabidopsis and rice. OsGIGANTEA (OsGI), Heading date 1 (Hd1), Heading date 3a (Hd3a), and Rice Flowering locus T 1 (RFT1) are the ortholog of Arabidopsis GI, CO, and FT in rice. Hd3a and RFT1 proteins are the florigens in rice [166,167,168]. OsGI, Hd1, and Hd3a form OsGI-Hd1-Hd3a module to control rice flowering. OsGI activates the expression of *Hd1* to promote synthesis of Hd3a and RFT1 to accelerate flowering [169,170,171,172].

*Early heading date 1* (*Ehd1*) encodes a plant specific B-type response regulator, which is activated by OsMADS51. Ehd1 activates the expression of *Hd3a* and *RFT1* to promote rice flowering [173]. Ghd7, OsMADS56, OsLEC1 and FUSCA-Like 1 (OsLFL1), and DTH8/Ghd8 repress the expression of *Ehd1* to delay flowering time of rice [174,175,176,177,178]. Ehd2, Ehd3, and Ehd4 activate the expression of *Ehd1*. Activated Ehd1 enhances the expression of *Hd3a* to promote flowering [179,180,181]. Interaction Protein 1 (SIP1), a C2H2 zinc finger domain-containing protein, binds to the promoter of *Ehd1* and recruits Trithorax-like protein, OsTrx1, to modify the chromatin with H3K4me3 mark and activates the expression of *Ehd1* to promote flowering of rice [182].

PRC2 is involved in regulation of flowering time. OsVIN3-Like (OsVIL) proteins are PHD domain containing protein, a homolog of Arabidopsis VIN3. There are four VIN3-like proteins, PHD proteins, in rice, termed OsVIL1 to OsVIL4 [183,184]. OsVIL2 and LC2/OsVIL3 have been reported to associate with PRC2 to regulate flowering time in rice [183,184,185]. The *osvil2* mutant and *OsVIL2*–RNAi lines display late flowering phenotype under both LD and SD conditions [184,185]. The expression of *OsLFL1* is increased, but the expression of *Ehd1*, *Hd3a,* and *RFT1* is reduced by *osvil2* mutation [185]. *OsLFL1* chromatin is modified by H3K27me3 mark. OsVIL2 physically interacts with OsEMF2b, a component of PRC2 in rice. OsVIL2 binds directly to *OsLFL1* chromatin to recruit PRC2 through interacting with OsEMF2b and catalyze H3K27 trimethylation on the locus of *OsLFL1*. H3K27me3 enrichment is diminished in the *osvil2* mutants. The *osemf2b* mutant flowers late by increasing *OsLFL1* expression and decreasing *Ehd1* expression [185]. These results suggest that OsVIL2, together with PRC2, represses *OsLFL1* expression and increases the expression of *Ehd1* to induce flowering [185].

LC2/OsVIL3 promote flowering of rice. The *lc2* mutation delays heading date by reducing the expression of *Hd1* and *Hd3a* under SD conditions. LC2 physically interacts with OsVIL2, but not with OsMSI1, OsCLF, OsEMF2a, OsEMF2b, OdFIE1, and OsFIE2. LC2 binds to the promoter region of a floral repressor *OsLF* and represses the *OsLF* expression via H3K27 trimethylation modification. *OsLF* directly regulates the *Hd1* expression through binding to *Hd1* promoter. OsLF, a bHLH transcription factor, interacts with OsPIL13 and OsPIL15 to repress rice flowering [186]. These results suggest that LC2 may bind to the promoter region of target genes to recruit PRC2 and modify its target genes [184].

## 6. Concluding Remarks

PRC2 and PRC1 play critical roles in maintaining the stable inactivated states of developmental genes by catalyzing H3K27me3 and H2AK119ub histone modifications, respectively. To ensure proper development, PcG proteins are required to precisely program gene expression in temporal- and spatial-dependent manners. The major components of PRC1 and PRC2 and their functions have extensively been explored in Arabidopsis and rice [6,7]. So far, the evidence suggests that both PRC1 and PRC2 are involved in the maintenance of stem cell activity and root development. After germination, multiple forms of PRC2s repress the expression of seed development genes, and PRC1 represses seed development and flower genes. EMF-PRC2, VRN-PRC2, and PRC1 control flowering time or floral transition by repressing the relative gene expression.

Though recent progresses have advanced our understanding of the PcG mechanisms on regulating major phase transitions in Arabidopsis and rice, more questions remain. It has been known that about 20% of the genes in Arabidopsis are modified by H3K27 tri-methylation [32,42,43]. Only small part of modified gene function is uncovered. Discovering the functions of modified genes is the challenges we are facing. For most of these identified targets, it is still not known precisely how and in which cells, the PcG system regulates gene expression. Because PcG proteins lack DNA-binding activity, the recruitment of PRC1 and PRC2 to their target chromatin is one of the major open questions in PcG-mediated gene silencing. Up to date, PRE *cis*-elements, transcription factors, DNA-binding proteins, and lncRNAs are proposed to contribute to the recruitment [6,187]. It is expected that various mechanisms exist, and further identifications of PcG-interacting factors would extend our understanding and allow us to extract fundamental principles underlying epigenetic regulation of eukaryote developmental programs.

Addressing these questions will further elucidate the functions of PcG proteins in silencing gene expression and regulating plant development.

## Figures and Tables

**Figure 1 ijms-21-05871-f001:**
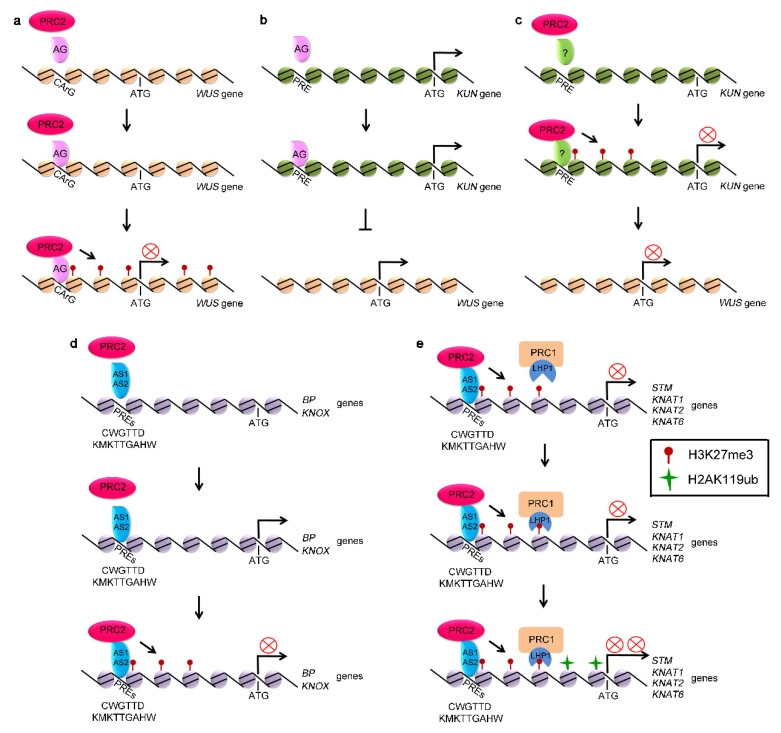
Polycomb-group (PcG) proteins repressed the expression of stem cell fate related genes. (**a**) polycomb repressive complex 2 (PRC2) silences the expression of WUSCHEL (*WUS)* through binding to the Polycomb Response Elements (PREs), such as CArG located in the promoter of *WUS* and generating H3K27me3 at *WUS* locus. (**b**) AG represses the expression of *WUS* through activating *KUN*, a suppressor of *WUS*. (**c**) PRC2 activates the expression of *WUS* through silencing *KUN*, competing with AG. (**d**) PRC2 represses the expression of *BP* and *KNOX* genes through binding to PREs, such as CWGTTD and KMKTTGAHW. (**e**) *KNOX* genes, including *STM*, *KNAT1*, *KNAT2*, and *KNAT6* genes, marked with H3K27me3 by PRC2.

**Table 1 ijms-21-05871-t001:** Components of polycomb repressive complex 1 (PRC1) and polycomb repressive complex 2 (PRC2) complexes discovered in *Drosophila*, Arabidopsis, and rice.

Complex	*Drosophila*	Arabidopsis	Rice	Function
PRC2	E(z)	CLF SWN MEA	OsCLF OsSET1	H3K27me3 HMT
	Su(z)12	EMF2 VRN2 FIS2	OsEMF2a OsEMF2b	Stimulates H3K27me3 HMT
	Esc	FIE	OsFIE1 OsFIE2	Stable and enhance E(z)
	N55	MSI1-5	OsMSI1	Bind to histones and Su(z)12
PRC1	dRING/Sce	AtRING1A AtRING1B	OsRING1	H2A monoubiquitination
	Psc	AtBMI1A AtBMI1B AtBMI1C	unknown	H2A monoubiquitination
	Pc	EMF1	OsEMF1a OsEMF1b	Chromatin compaction
		LHP1	OsLHP1	Binds to H3K27me3 mark
		VRN1	unknown	Recruiter, interacts with AtBMI1A/1B/1C
		VAL1/2/3	unknown	Recruiter, interacts with AtBMI1A/1B/1C
	Ph	unknown	unknown	Unknown

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
