# Peer review of "Knowing When to Silence: Roles of Polycomb-Group Proteins in SAM Maintenance, Root Development, and Developmental Phase Transition"

_ijms, 2020, doi:10.3390/ijms21165871_

Round 1

Reviewer 1 Report

This review paper focuses on recent discoveries about the regulatory role of PcG proteins in plant development. It seems that the papers that have been reported until recently have been well organized and focused. Therefore, it is judged that it can be sufficiently posted on IJMS.

Author Response

Reviewer 1

This review paper focuses on recent discoveries about the regulatory role of PcG proteins in plant development. It seems that the papers that have been reported until recently have been well organized and focused. Therefore, it is judged that it can be sufficiently posted on IJMS.

Our response: It is a good suggestion. We reorganized the manuscript and added latest published paper to the revised manuscript.

Reviewer 2 Report

The review “Knowing When to Silence: Roles of Polycomb Repressive Complexes in Regulating Plant Development” written by Yang and colleagues described the role of PcG in regulating plant development through repressing gene expression. Authors organize their review focusing on different aspects of plant development trying to group the knowledge on the topic discovered in the two model species Arabidopsis thaliana and Oryza sativa. In the introduction, authors report the known PcG in Arabidopsis and rice and their correlation with the Drosophila, PRC core components. The review is divided into three main paragraphs, the first describing the roles of PRC2 proteins in SAM maintenance and differentiation where authors concentrated their effort on shoot apical meristem. The second report the roles of PcG proteins in embryo to seedling phase transition and give an account of regulation by PRCs in seed development. Then the focus is moved to PcG proteins in floral transition in which an overview of the regulation of main players through PRC1 and PRC2 complexes.

I understand that the argument the authors selected is enormous and the number of information available are difficult to compress in a review, but in this form, in my opinion, the result is not complete and does not meet the expectations set out in the title. The term “Plant development” is too generic, the authors should focus on a specific phase in plant development and describe for the selected argument the latest finding. For example, root development is completely ignored (doi: 10.1016/j.molcel.2019.12.015, doi: 10.1038/s41598-018-26349-w, doi: 10.1093/pcp/pcx006 some example of involvement of PcG in root development). Also, flower development is not taken into consideration (doi: 10.1111/tpj.14673, doi: 10.1104/pp.17.01269, doi: 10.1371/journal.pgen.1005771 some example of the involvement of PcG in flower development)

I would suggest narrowing the developmental phases considered adjusting the title.

In the introduction, a table summarizing all the components discovered in Arabidopsis, rice and Drosophila with relative references will help a lot the readers to get a complete and quick overview of the different complexes and the “evolution” across the organisms.

Paragraph 2.1 is too long and it should be shortened in my opinion.

It would also be interesting to present and discuss the latest finding on molecular mechanisms of action of PRCs complexes and the results of the High-throughput data obtained from different species and various PRCs component or histone modifications concerning developmental processes. In doi: 10.1038/ng.3937, for example, the authors deepen our knowledge on the PRE sequences in Arabidopsis and it would be worth to mention it in the review.

A general suggestion is to revise the language. There are some errors and phrases difficult to understand and to read fluently (For example line 19 plural in discovery, 25 are initially -> have been initially, 34-35 wrong tenses, from 213 to 239, difficult to read, some errors, revise completely the paragraph, 353 error etc…)

In general, I would suggest adding the latest literature on the topic since in the present form the manuscript present few details that have not been already discussed in other recent reviews on plant development.

Author Response

Reviewer 2

The review “Knowing When to Silence: Roles of Polycomb Repressive Complexes in Regulating Plant Development” written by Yang and colleagues described the role of PcG in regulating plant development through repressing gene expression. Authors organize their review focusing on different aspects of plant development trying to group the knowledge on the topic discovered in the two model species Arabidopsis thaliana and Oryza sativa. In the introduction, authors report the known PcG in Arabidopsis and rice and their correlation with the Drosophila, PRC core components. The review is divided into three main paragraphs, the first describing the roles of PRC2 proteins in SAM maintenance and differentiation where authors concentrated their effort on shoot apical meristem. The second report the roles of PcG proteins in embryo to seedling phase transition and give an account of regulation by PRCs in seed development. Then the focus is moved to PcG proteins in floral transition in which an overview of the regulation of main players through PRC1 and PRC2 complexes.

Q1: I understand that the argument the authors selected is enormous and the number of information available are difficult to compress in a review, but in this form, in my opinion, the result is not complete and does not meet the expectations set out in the title. The term “Plant development” is too generic, the authors should focus on a specific phase in plant development and describe for the selected argument the latest finding. For example, root development is completely ignored (doi: 10.1016/j.molcel.2019.12.015, doi: 10.1038/s41598-018-26349-w, doi: 10.1093/pcp/pcx006 some example of involvement of PcG in root development). Also, flower development is not taken into consideration (doi: 10.1111/tpj.14673, doi: 10.1104/pp.17.01269, doi: 10.1371/journal.pgen.1005771 some example of the involvement of PcG in flower development). I would suggest narrowing the developmental phases considered adjusting the title.

Our response: It is a good suggestion. We changed the title and added several topic to the revised manuscript according to reviewers’ suggestion.

Q2: In the introduction, a table summarizing all the components discovered in Arabidopsis, rice and Drosophila with relative references will help a lot the readers to get a complete and quick overview of the different complexes and the “evolution” across the organisms.

Our response: It is good suggestion. Table 1 is added to the revised manuscript to summarize the components discovered in Drosophila Arabidopsis, and rice.

Q3: Paragraph 2.1 is too long and it should be shortened in my opinion.

Our response: It is a good suggestion. This section is shortened.

Q4: It would also be interesting to present and discuss the latest finding on molecular mechanisms of action of PRCs complexes and the results of the High-throughput data obtained from different species and various PRCs component or histone modifications concerning developmental processes. In doi: 10.1038/ng.3937, for example, the authors deepen our knowledge on the PRE sequences in Arabidopsis and it would be worth to mention it in the review.

Our response: It is a good suggestion. We added this in the revised manuscript.

Q5: A general suggestion is to revise the language. There are some errors and phrases difficult to understand and to read fluently (For example line 19 plural in discovery, 25 are initially -> have been initially, 34-35 wrong tenses, from 213 to 239, difficult to read, some errors, revise completely the paragraph, 353 error etc…)

Our response: It is a good suggestion. The mistakes were corrected in the revised manuscript.

Q6: In general, I would suggest adding the latest literature on the topic since in the present form the manuscript present few details that have not been already discussed in other recent reviews on plant development.

Our response: It is a good suggestion. The latest published papers were cited in the revised manuscript.

Round 2

Reviewer 2 Report

The new version of the manuscript is quite improved, the authors follow basically all my suggestion. Nevertheless, I suggest a careful revision of the English:

-line 54: have been identified

-lines 72-74: rephrase, not clear

-line 136 have been recently reported

-line 188: the findings are consistent

-line 189: revise the phrase "to" seems to be an error

-lines 230-231: revise the phrase, not clear

-lines 368-370: delete

In paragraph 4.1 I would add the ref doi: 10.1105/tpc.19.00764 and describe briefly the role of BPC and FIS-PRC2 (FERTILIZATION-INDEPENDENT SEED-Polycomb Repressive Complex2), which represses FUS3 in the endosperm during early seed development.

Minor points:

-line 144: "Mutations in as1 or as2 

" genes in uppercase.

-line 148: "Mutations of AtRing1a and AtRing1b" genes in uppercase.

Moreover in doi: 10.1111/tpj.14673 the role of LHP1 and BPCs in the repression of STK in IM is described, which could be added in paragraph 2.

Author Response

Review 2’ comments:

The new version of the manuscript is quite improved, the authors follow basically all my suggestion. Nevertheless, I suggest a careful revision of the English:

Q1: -line 54: have been identified

Our response: We change “In rice (Oryza sativa), two E(z)-like proteins in the rice genome are identified termed……” in line 55 to “In rice (Oryza sativa), two E(z)-like proteins in the rice genome have been identified termed……”.

Q2: -lines 72-74: rephrase, not clear

Our response: We change “In addition to the involvement of LHP1 in PRC1 complex, LHP1 is also associated with PRC2 to deposit H3K27me3 mark on the chromatin through interacting with MSI1 and EMF2” in lines 72-74 to “In addition to the involvement in PRC1 complex, LHP1 is also associated with PRC2 through interacting with MSI1 and EMF2 to deposit H3K27me3 mark on the chromatin”

Q3: -line 136 have been recently reported

Our response: We change “It has been known that H3K27 trimethylation……” in line 137 to “It has been recently reported that H3K27 trimethylation……”.

Q4: -line 188: the findings are consistent

Our response: We change “These findings are consistence with……” in line 195 to “The findings are consistence with……”.

Q5: -line 189: revise the phrase "to" seems to be an error

Our response: We change “R-loops are able to be formed at many PREs in Drosophila embryos to correlate with repressive states of the target genes.” in lines 196-197 to “R-loops are able to be formed at many PREs in Drosophila embryos correlated with repressive states of the target genes.”

Q6: -lines 230-231: revise the phrase, not clear

Our response: We change “In addition, PRC1 is involved in seed development and seedling growth. Genes required for seed development are activated by emf1 mutants.” in lines 230-231 to “PRC1 is involved in seed development and seedling growth. Genes required for seed development are activated by emf1 mutation.”

Q7: -lines 368-370: delete

Our response: “Rice is a model system of monocots and an important cereal crop. The functions of PRC1 and PRC2 in development, yield formation, and quality control in rice remain far from clear.” in lines 368-370 is deleted.

Q8: In paragraph 4.1 I would add the ref doi: 10.1105/tpc.19.00764 and describe briefly the role of BPC and FIS-PRC2 (FERTILIZATION-INDEPENDENT SEED-Polycomb Repressive Complex2), which represses FUS3 in the endosperm during early seed development.

Our response: It is a good suggestion. We added it to 4.1.

Minor points:

Q9: -line 144: "Mutations in as1 or as2 " genes in uppercase.

Our response: We change "Mutations in as1 or as2 " in line 149 to "Mutations in AS1 or AS2 ".

Q10: -line 148: "Mutations of AtRing1a and AtRing1b" genes in uppercase.

Our response: We change "Mutations of AtRing1a and AtRing1b" in lines 153-154 to "Mutations of AtRING1A and AtRING1B".

Q11: Moreover in doi: 10.1111/tpj.14673 the role of LHP1 and BPCs in the repression of STK in IM is described, which could be added in paragraph 2.

Our response: It is a good suggestion. We added it to 4.1.